# Anti-Inflammatory Effects of *Berchemia floribunda* in LPS-Stimulated RAW264.7 Cells through Regulation of NF-κB and MAPKs Signaling Pathway

**DOI:** 10.3390/plants10030586

**Published:** 2021-03-19

**Authors:** Hyun Ji Eo, Jun Hyuk Jang, Gwang Hun Park

**Affiliations:** Forest Medicinal Resources Research Center, National Institute of Forest Science, Yeongju 36040, Korea; ehj56@naver.com (H.J.E.); wnseldu123@gmail.com (J.H.J.)

**Keywords:** anti-inflammatory effect, *Berchemia floribunda*, MAPKs, NF-κB

## Abstract

*Berchemia floribunda* (Wall.) Brongn. (BF), which belongs to Rhamnaceae, is a special plant of Anmyeon Island in Korea. BF has been reported to have antioxidant and whitening effects. However, the anti-inflammatory activity of BR has not been elucidated. In this study, we evaluated the anti-inflammatory effect of leaves (BR-L), branches (BR-B) and fruit (BR-F) extracted with 70% ethanol of BR and elucidated the potential signaling pathway in LPS-induced RAW264.7 cells. BR-L showed a strong anti-inflammatory activity through the inhibition of NO production. BR-L significantly suppressed the production of the pro-inflammatory mediators such as iNOS, COX-2, IL-1β, IL-6 and TNF-α in LPS-stimulated RAW264.7 cells. BR-L suppressed the degradation and phosphorylation of IκB-α, which contributed to the inhibition of p65 nuclear accumulation and NF-κB activation. BR-L obstructed the phosphorylation of MAPKs (ERK1/2, p38 and JNK) in LPS-stimulated RAW264.7 cells. Consequently, these results suggest that BR-L may have great potential for the development of anti-inflammatory drugs to treat acute and chronic inflammatory disorders.

## 1. Introduction

Inflammation is one of the defensive responses of living tissues to toxins, physical damage and external chemical stimuli. Inflammation is the result of the immune reaction of harmful stimuli such as irritants and pathogens, and activated macrophages such as RAW 264.7 cells are involved in the process of inflammation [1,2]. Activated macrophages produce increased expression of the inflammatory mediators such as nitric oxide (NO), prostaglandins 2 (PGE2), inducible nitric oxide synthase (iNOS), cyclooxygenase-2 (COX-2), interlukin-1 beta (IL-1β), IL-6 and tumor necrosis factor alpha (TNF-α). The nuclear factor kappa B (NF-κB) signaling pathway plays a central role, releasing inflammatory cytokines in lipopolysaccharides (LPS)-induced RAW 264.7 cells. The mitogen-activated protein kinases (MAPKs) signaling pathway is associated with inflammatory responses including cell survival, apoptosis, mitosis and differentiation. Thus, the suppression of the NF-κB and MAPKs signaling pathway has been regarded as one of the pharmacological approaches to human diseases such as cancer, arthritis, alzheimer’s disease, asthma allergies and inflammatory diseases [3,4,5]. Recently, many researchers are interested in various medicinal plants for the evaluation of anti-cancer, anti-oxidant and anti-inflammatory phytochemicals such as phenols, flavonoids and tannins, which have received more attention for their potential role in the prevention or treatment of human diseases [6,7,8].

*Berchemia floribunda* (Wall.) Brongn. (BF), which is one species of the genus Berchemia (Rhamnaceae family), is distributed in Korea, China and Japan, but in Korea it can be only found on the Anmyeon Island. BF has been reported to have antioxidant, whitening and anti-cancer effects [9,10,11,12,13]. The phytochemicals are chemicals produced by plants through primary or secondary metabolites. These compounds exhibit pharmacological effects applicable to the treatment of various human diseases, such as diabetes, cancer, and obesity [14,15,16,17]. Several major compounds have been isolated and identified from the roots of BF, such as anthraquinones [18,19]. Although the phytochemical content of the roots of BF has been extensively investigated, information about the chemical compounds of other parts of this plant is still sparse. The aim of this study was to investigate the anti-inflammatory effect of BF through the inhibition of NF-κB and MAPK signaling activation in LPS-induced RAW264.7 cells.

## 2. Materials and Methods

### 2.1. Chemical Reagents

3-(4,5-dimethylthizaol-2-yl)-2,5-diphenyl tetrazolium bromide (MTT) and Lipopolysaccharide (LPS) were purchased in Sigma Aldrich (St. Louis, MO, USA). Antibodies against phospho-IκB-α (#2859), IκB-α (#4814), phospho-p38 (p-p38) (#4511), total-p38 (#9212), phospho-extracellular signal-regulated kinase1/2 (p-ERK1/2) (#4377), ERK1/2 (#9102), phospho- c-Jun N-terminal kinase (p-JNK) (#9251), JNK (#9258) and β-actin were purchased in Cell Signaling (Bervely, MA, USA).

### 2.2. Preparation of the Extracts of Leaves, Branches and Fruit from BF

BF was collected in 2018 from Anmyeon-island, Chungcheongnam-do, Korea, and formally identified by Ho Jun Son as a researcher of Forest Medicinal Resources Research Center, Korea (voucher specimen: FMCBrAMD-1807-1-3). A total of 20 g of leaves, branches, and fruit of BF were extracted with 400 mL of 70% ethanol for 48 h under stirring at room temperature. After 48 h, the ethanol extracts were filtered and concentrated using a vacuum evaporator at 40 °C (N-1000, EYELA, Tokyo, Japan) and then freeze-dried.

### 2.3. Cell Culture and Treatment

RAW264.7 macrophages were purchased American Type Culture Collection (ATCC, Virginia, USA) and grown in DMEM/F-12 supplemented with 10% fetal bovine serum (FBS), penicillin (100 units/mL) and streptomycin (100 μg/mL) under a humidified atmosphere of 5% CO_2_ at 37 °C. BF was dissolved in dimethyl sulfoxide (DMSO) and treated to cells. DMSO was used as a vehicle and the final DMSO concentration did not exceed 0.1% (*v*/*v*).

### 2.4. Measurement of Cell Viability

Cell viability was measured using an MTT assay system. RAW264.7 cells were cultured at a concentration of 1 × 10^5^ cells/well in 12-well culture plates. The cells were treated with BF for 24 h. Then, the cells were incubated with 200 μL of MTT solution (1 mg/mL) for an additional 2 h. The resulting crystals were dissolved in dimethyl sulfoxide (DMSO). The formation of formazan was measured by reading absorbance at a wavelength of 570 nm (Perkin Elmer, Waltham, MA, USA).

### 2.5. NO Inhibition Activity

RAW264.7 cells were plated in a 12-well plate overnight. Cells were pretreated with each part (leaves, branches and fruit) of BF at the indicated concentrations for 2 h, and then co-treated with LPS (1 μg/mL) for an additional 18 h. After that, 50 μL of the media was mixed with 50 μL of Griess reagent (Sigma Aldrich, St. Louis, MO, USA). The mixture was incubated for an additional 5 min at room temperature and the absorbance was measured at 540 nm (Perkin Elmer, Waltham, MA, USA).

### 2.6. Isolation of Nucleus Fraction

Nuclear fractions from RAW264.7 cells after treatment of BF-L and LPS were prepared using a nuclear extract kit (Active Motif, Carlsbad, CA, USA) according to the manufacturer’s protocols. Briefly, RAW264.7 cells were harvested with cold 1 × hypotonic buffer and reacted at 4 °C for 15 min. Then, detergent was added and vortexed for 10 s. RAW264.7 cells were centrifuged at 15,000 rpm for 10 min at 4 °C and the cell pellets were collected for the extraction of nuclear fraction. The collected pellets were extracted using lysis buffer through incubation at 4 °C for 30 min under shaking. After 30 min, nuclear fractions from the cell pellets were centrifuged at 15,000 rpm for 10 min at 4 °C, and the supernatants (nuclear fraction) were stored at −80 °C for further analysis.

### 2.7. Reverse Transcriptase-Polymerase Chain Reaction (RT-PCR)

Total RNA was prepared using a RNeasy Mini Kit (Qiagen, Valencia, CA, USA) and total RNA (1 µg) was reverse transcribed using a Verso cDNA kit (Thermo Scientific, Pittsburgh, PA, USA) according to the manufacturer’s protocol for cDNA synthesis. PCR was carried out using PCR Master Mix Kit (Promega, Madison, WI, USA) with primers for mouse iNOS, COX-2, IL-1β, TNF-α and mouse GAPDH as follows: mouse iNOS: forward 5′-gtgcgcctctggtcttgcaagc-3′ and reverse 5′-aggggcaggctgggaattcg-3′, mouse COX-2: forward 5′-ggagagactatcaagatagtgatc-3′ and reverse 5′-atgtgcagtagacttttacagctc-3′, mouse IL-1β: forward 5′-gaagctgtg gcagctacctatgtct-3′ and reverse 5′-ctctgcttgtgaggtgctgatgtac-3′, mouse TNF-α: forward 5′-tactgaacttcggggtgattggtcc-3′ and reverse 5′-cagccttgtcccttgaagagaacc-3′, mouse GAPDH: forward 5′-caggagcgagaccccactaacat-3′ and reverse 5′-gtcagatccacgacggacacatt-3′.

### 2.8. Sodium Dodecyl Sulphate–Polyacrylamide Gel Electrophoresis (SDS-PAGE) and Western Blot

Cells were washed with 1 × PBS, and lysed in radioimmunoprecipitation assay (RIPA) buffer (Boston Bio Products, Ashland, MA, USA) supplemented with protease inhibitor cocktail (Sigma Aldrich) and phosphatase inhibitor cocktail (Sigma Aldrich), and centrifuged at 15,000 rpm for 10 min at 4 °C. Protein concentration was determined by the bicinchoninic acid (BCA) protein assay (Pierce, Rockford, IL, USA) using bovine serum albumin (BSA) as the standard. The proteins were separated on SDS-PAGE and transferred to nitrocellulose membrane. The membranes were blocked for non-specific binding with 5% skim milk in Tris-buffered saline containing 0.05% Tween 20 (TBS-T) for 1 h at room temperature, and then incubated with specific primary antibodies in 5% skim milk at 4 °C overnight. After three washes with TBS-T, the blots were incubated with horse radish peroxidase (HRP)-conjugated immunoglobulin G (IgG) for 1 h at room temperature and chemiluminescence was detected with an ECL Western blotting substrate and visualized in chemi doc (Bio-rad, Chemi Doc MP Imaging system, Hercules, CA, USA).

### 2.9. Statistical Analysis

All the data are shown as mean ± SD (standard deviation). Statistical analysis was performed with one-way ANOVA followed by Dunnett’s test. Differences with * *p* or # *p* < 0.05 were considered statistically significant.

## 3. Results

### 3.1. Effect of BF-L on NO Production and iNOS, COX-2, IL-β, IL-6 and TNF-α Expression in LPS-Stimulated RAW264.7 Cells

Nitric oxide (NO) and reactive oxygen species exert multiple modulating effects on inflammation and play an important role in the controlling of immune responses. They affect advanced inflammation at almost every stage [20]. We used the mouse macrophage cell line, RAW 264.7 cells for evaluating the anti-inflammatory effects of the BF. To determine whether BF reduces NO generation by LPS, RAW 264.7cells were pretreated with the extracts from the BF for 2 h and then co-treated with LPS (1 μg/mL) for an additional 24 h. After stimulation, the cell medium was collected and the production of NO was measured using the Griess reagent. As a result, ethanol extracts of BF leaves (BF-L), branches (BF-B) and fruit (BF-F) each showed an NO production of about 32.9%, 60.3% and 94.9% at 100 μg/mL, respectively (Figure 1). The effects of BF on the inhibition of NO production in LPS-induced RAW264.7 cells were examined. As shown in Figure 2a, BF-L concentration-dependently (0, 12.5, 25, 50, 100 μg/mL) reduced the excessive production of NO in LPS-induced RAW264.7 cells. MTT assay showed that BF-L was not toxic to RAW264.7 cells (Figure 2b). As shown in Figure 2c,d, LPS-mediated overexpression of iNOS, COX-2, IL-1β, IL-6 and TNF-α was slightly decreased by BF-L in LPS-stimulated RAW264.7 cells.

### 3.2. Effect of BF-L on NF-κB Signaling Activation in LPS-Stimulated RAW264.7 Cells

The NF-κB pathway does indeed regulate pro-inflammatory cytokine production, leukocyte recruitment, or cell survival, which are key contributors to the inflammatory response [21]. Therefore, in order to examine the effect of BF-L on the inhibition of NF-κB activity, as shown in Figure 3a, the treatment of BF-L in LPS-induced RAW264.7 cells suppressed the phosphorylation and degradation of IκB in a concentration-dependent manner (0, 12.5, 25, 50 and 100 μg/mL). As shown in Figure 3b, it was confirmed that BF-L inhibited nuclear translocation of p65. These results suggest that BF-L has an anti-inflammatory effect by inhibiting NF-κB activity.

### 3.3. Effect of BF-L on MAPKs Signaling Activation in LPS-Stimulated RAW264.7 Cells

There are three major types of MAPK: extracellular signal-regulated kinase (ERK), c-Jun N-terminal kinase (JNK), and p38 MAPK. Phosphorylation of the MAPKs is associated with inflammatory responses including mitosis, differentiation, cell survival and cell apoptosis [22,23,24]. We investigated whether BF-L inhibits the phosphorylation of ERK1/2, p38 or JNK to elucidate the inhibitory effect of BF-L against MAPKs signaling activation in LPS- induced RAW264.7 cells. As shown in Figure 4a,b, the treatment of LPS alone phosphorylated ERK1/2, p38 and JNK compared with the cells without LPS. However, the presence of BF-L inhibited LPS-mediated phosphorylation of ERK1, p38 and JNK in LPS-stimulated RAW264.7 cells. These results suggest that BF-L has anti-inflammatory effects by inhibiting MAPK activity.

## 4. Discussion

Under LPS-stimulation, RAW264.7 cells generated NO production, which plays an important role in inflammatory response [25,26,27]. There was a BF-L-induced decrease in NO production in LPS-stimulated RAW264.7 cells. Increased pro-inflammatory cytokines such as TNF-α, IL-6, and IL-1β play an important role in the development of early inflammatory responses [28]. The NF-κB pathways produce excessive inflammatory mediators such as NO, iNOS, COX-2 and IL-1β. NF-κB is one of the most important transcription factors involved in inflammatory mechanisms in immune cells. Abnormal NF-κB activity due to various causes has been known as a mechanism of various autoimmune diseases such as atopy, allergy, arthritis [29,30,31]. BF-L inhibited inflammatory mediator gene expression such as iNOS, COX-2, IL-1β, IL-6 and TNF-α, which are known to be involved in the pathogenesis of inflammatory diseases. The MAPK signaling cascade plays an essential role in the initiation of inflammatory responses [32]. MAPK pathway is involved in the modulation of iNOS as well as the expression of various cytokines in LPS-stimulated RAW 264.7 cells [33]. We found that inhibition of ERK1/2, p38 and JNK phosphorylation may be involved in the anti-inflammatory effect of BF-L. These results suggest that the inhibition of NF-KB and MAPK pathway is a potential mechanism by which BF-L exerts anti-inflammatory effects.

## 5. Conclusions

In this study, we demonstrated that leaves of *Berchemia floribunda* extracts (BF-L) showed anti-inflammatory effect through attenuating the generation of the pro-inflammatory mediators such as NO, iNOS, COX-2, IL-1β, IL-6 and TNF-α via the inhibition of NF-κB and MAPK signaling activation. For the first time, the anti-inflammatory effect of BF-L against NF-κB signaling and MAPK pathways was investigated in RAW264.7 cells. These findings suggest that BF-L may have great potential for the development of anti-inflammatory drugs to treat acute and chronic inflammatory disorders. However, the anti-inflammatory effect of BF-L in vivo and the identification of major compounds from BF-L with the anti-inflammatory effect needs further study.

## Figures and Tables

**Figure 1 plants-10-00586-f001:**
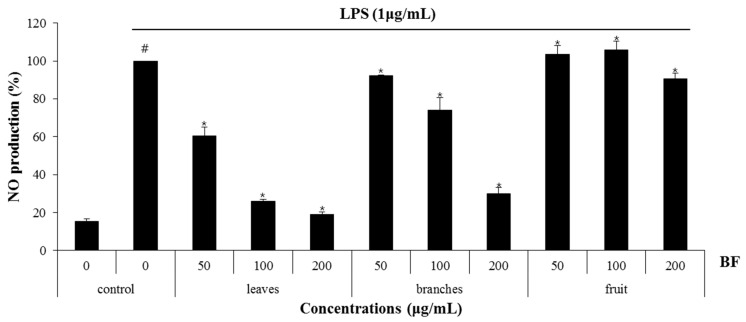
The effect of leaves, branches, and fruit of *Berchemia floribunda* (BF) on nitric oxide (NO) production in lipopolysaccharides (LPS)-stimulated RAW264.7 cells. RAW264.7 cells were pretreated with leaves, branches and fruits of BF for 6 h and then co-treated with LPS (1 μg/mL) for 18 h. NO production was measured by Griess assay. # *p* < 0.05 compared to the cells without the treatment, and * *p* < 0.05 compared to the cells treated with LPS alone.

**Figure 2 plants-10-00586-f002:**
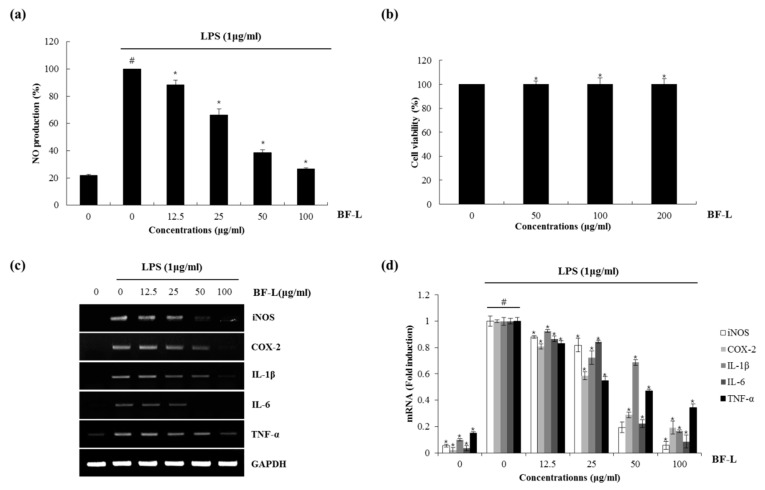
The effect of BF-L on NO production and iNOS, COX-2, IL-1β, IL-6 and TNF-α expression in LPS-stimulated RAW264.7 cells. (**a**) RAW264.7 cells were pre-treated with BF-L at the indicated concentrations for 6 h and then co-treated with LPS (1 μg/mL) for an additional 18 h. After treatment, NO production was measured using the media and Griess reagent. (**b**) RAW264.7 cells were treated with BF-L at the indicated concentrations for 24 h. Cell viability was measured using MTT assay system and expressed as % cell viability. (**c**,**d**) For RT-PCR, RAW264.7 cells were pre-treated with BF-L at the indicated concentrations for 6 h and then co-treated with LPS (1 μg/mL) for an additional 18 h. Total RNA was isolated and RT-PCR was performed for iNOS, COX-2, IL-1β, IL-6 and TNF-α values given are the mean ± SD (n = 3). * *p* < 0.05 compared to LPS treatment without BF-L. # *p* < 0.05 compared to the cells without BF-L, and * *p* < 0.05 compared to the cells treated with LPS alone. GAPDH was used as an internal control for RT-PCR.

**Figure 3 plants-10-00586-f003:**
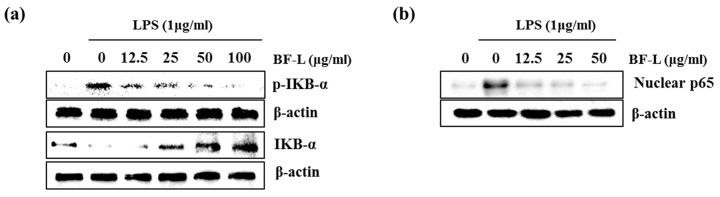
Effect of BF-L on NF-κB signaling activation in LPS-stimulated RAW264.7 cells. (**a**) RAW264.7 cells were pretreated with BF-L for 6 h and then co-treated with LPS (1 μg/mL) for 30 min. For Western blot analysis, the cell lysates were subjected to SDS-PAGE and the Western blot was performed using antibodies against p-IκB-α and IκB-α. β-actin was used as internal control. (**b**) RAW264.7 cells were pretreated with BR-L for 6 h and then co-treated with LPS (1 μg/mL) for 30 min. After the treatment, the nucleus fraction was prepared. For Western blot analysis, the cell lysates were subjected to SDS-PAGE and the Western blot was performed using antibodies against p65. Actin was used as internal control for Western blot analysis.

**Figure 4 plants-10-00586-f004:**
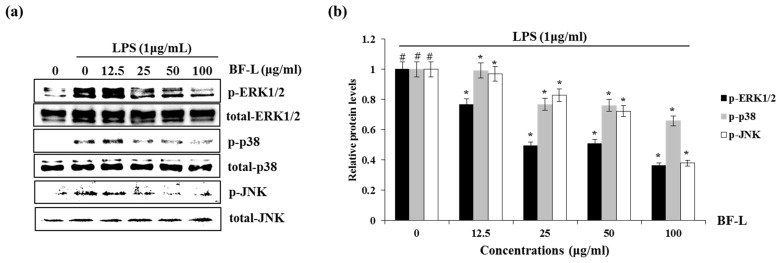
Effect of BF-L on MAPKssignaling activation in LPS-stimulated RAW264.7 cells. (**a**) and (**b**) RAW264.7 cells were pretreated with BF-L for 6 h and then co-treated with LPS (1 μg/mL) for 30 min. For Western blot analysis, the cell lysates were subjected to SDS-PAGE and the Western blot was performed using antibodies against p-ERK1/2, p-p38, p-JNK, total ERK1/2, total p38 and total JNK. # *p* < 0.05 compared to the cells without BF-L, and * *p* < 0.05 compared to the cells treated with LPS alone.

## Data Availability

Data available in a publicly accessible repository.

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
