# Peer review of "Anti-Inflammatory Effects of Berchemia floribunda in LPS-Stimulated RAW264.7 Cells through Regulation of NF-κB and MAPKs Signaling Pathway"

_plants, 2021, doi:10.3390/plants10030586_

Round 1

Reviewer 1 Report

The authors examined the influence of Berchemia floribunda extract on the development of inflammatory responses by using LPS-stimulated macrophage cell line, RAW264.7 and an in vitro cell culture technique. Although the data are very worth, several additional experiments are required for publication in this journal.

Major comment

The authors should examine the most important component(s) (or the most effective component) showing the suppressive effects on the production of nitric oxide and inflammatory cytokines.

Minor comments

  1. The authors should explain the methods used for the activation of NF-kappaB and MAPKs in the section of Materials and Methods.

    2. The authors should examine the protein levels of inflammatory cytokines, such as IL-1b, IL-6 and TNF-a, in addition to mRNA levels.

Author Response

1. The authors should explain the methods used for the activation of NF-kappaB and MAPKs in the section of Materials and Methods.
A: we explained the SDS-PAGE and western blot in the section of Materials and Methods.
In addition, activation of NF-kappaB and MAPKs show results in fig. 2-3.

2. The authors should examine the protein levels of inflammatory cytokines, such as IL-1b, IL-6 and TNF-a, in addition to mRNA levels.
A: We confirmed that cox-2 decreases in protein levels. However, it was not shown in the results of this study. Thank you.

Reviewer 2 Report

The paper “Anti-Inflammatory Effect and Phytochemical Analysis of Berchemia floribunda Extract” reports the anti-inflammatory effect of BF leaves, branches and fruit extracts and the potential signaling pathway in macrophages.

Some points should be clarified before publication:

- In the abstract, the major bioactive compounds detected in the extract(s) by UHPLC-TOF-HRMS should be reported.

- The introduction should be implemented. The renewed interest in the use of plant extracts for the treatment of human diseases could be better discussed, also in view of their innovative administration by delivery systems: see for example “Hura crepitans L. Extract: Phytochemical Characterization, Antioxidant Activity, and Nanoformulation, Pharmaceutics 2020, 12, 553; Phytochemical Profile of Capsicum annuum L. cv Senise, Incorporation into Liposomes, and Evaluation of Cellular Antioxidant Activity, Antioxidants 2020, 9, 428”.

- The three BF extracts were tested in cells for NO production inhibition. Then, the Authors tested only BF leaves extract for COX-2, IL-1β, IL-6 and TNF-α expression, and NF-κB inhibition. Why? Moreover, also the phytochemical screening was assessed for BF leaves extract only.

- The discussion should be implemented with comparison with literature data.

Author Response

- The introduction should be implemented. The renewed interest in the use of plant extracts for the treatment of human diseases could be better discussed, also in view of their innovative administration by delivery systems: see for example “Hura crepitans L. Extract: Phytochemical Characterization, Antioxidant Activity, and Nanoformulation, Pharmaceutics 2020, 12, 553; Phytochemical Profile of Capsicum annuum L. cv Senise, Incorporation into Liposomes, and Evaluation of Cellular Antioxidant Activity, Antioxidants 2020, 9, 428”.
A: We added introduction about the treatment of human disease and phytochemicals of BF extracts. 
- The three BF extracts were tested in cells for NO production inhibition. Then, the Authors tested only BF leaves extract for COX-2, IL-1β, IL-6 and TNF-α expression, and NF-κB inhibition. Why? Moreover, also the phytochemical screening was assessed for BF leaves extract only.
A: NO production inhibition(%) resulted in the highest activity in the leaves. So, we performed in vitro study to evaluate anti-inflammatory effects of BF-leaves and then elucidate the potential mechanisms. 
- The discussion should be implemented with comparison with literature data.
A: We checked and modified it.

Reviewer 3 Report

The manuscript presents data on the elucidation of the anti-inflammatory activity of extracts obtained from leaves, branches, and fruits of Berchemia floribunda (Wall.) Brongn. (BF), endemic to Anmyeon Island in South Korea, which antioxidant properties are escribed to. In the presented studies the anti-inflammatory activity was elucidated by the means of laboratory methods for the first time. The rationale for the establishment of this work ascertains its novelty and the importance in the context of the medicinal properties of the tested plant and potential application in drugs of the plant origin.

The drawback of the study seems the phytochemical analysis limited to only one, the most active extract obtained from leaves. As the extracts of branches displayed also some anti-inflammatory activity when the fruit extract was inactive, a comparison of the phytochemical profile and the content of individual compounds in all tested extracts could be interesting from a scientific point of view. Consequently,  it could be more clearly stated which compounds could determine the high activity of one specific extract, and are present in significant amounts in the active fractions or are absent in the inactive extracts. As the plant material shows great phytochemical variability it would be more interesting than referring to literature data.  

Minor Essential Revision:

LInes 27-35 There are some substantive repetitions in this part of the manuscript which should be corrected. 

Lines41-42 The sentence “is a plant in the Rhamnaceae, which is distributed in Korea and Japan, and is a type of species only on the Anmyeon-do in Korea.” is not clear. Is the plant widely distributed in Korea or endemic do Anmyeon?

English language level of the write up seems irregular in some text fragments, but the Reviewers isn’t able to qualify the linguistic dimension of the presented work.  

Author Response

-Lines41-42 The sentence “is a plant in the Rhamnaceae, which is distributed in Korea and Japan, and is a type of species only on the Anmyeon-do in Korea.” is not clear. Is the plant widely distributed in Korea or endemic do Anmyeon?
A: Sorry, we checked and modified it. BF is found in Anmyeon-do in Korea. 
Revision: Berchemia floribunda (Wall.) Brongn. (BF), one species of the genus Berchemia (Rhamnaceae family), was distributed in Korea, China and Japan.

-English language level of the write up seems irregular in some text fragments, but the Reviewers isn’t able to qualify the linguistic dimension of the presented work.  
A: If necessary, we will get to the English-editing service.

Reviewer 4 Report

The paper titled “Anti-Inflammatory Effect and Phytochemical Analysis of Ber chemia floribunda Extract” was well formulated in the study part of the anti-inflammatory activity but there are several aspects to consider before accepting the manuscript.

-The discussion is a bit poor and should be implemented; the authors confirmed that BF-L inhibited the gene expression of inflammatory mediators known to be involved in the pathogenesis of inflammatory diseases.

What is not very clear is the role played by compounds identified by UHPLC-TOF-HRMS as no assays have been performed with these molecules. In this regard, I suggest reformulating the final part of the discussion (lines 229-234).

-The "Conclusion" part needs to be rewritten as there are several errors (lines 236-238).

-Through HPLC-TOF-HRMS analysis, the authors report a list of compounds identified but did not provide quantitative information. Please add these data.

What percentage of the individual compounds are present in the analyzed matrix?

-The ethanolic extracts were investigated only by HPLC while it would have been very interesting to also evaluate the volatile composition by GC / MS to identify other molecules bioactive able to exert the anti-inflammatory effect in a synergistic way.  Why didn't the authors take this into consideration?

Author Response

-The discussion is a bit poor and should be implemented; the authors confirmed that BF-L inhibited the gene expression of inflammatory mediators known to be involved in the pathogenesis of inflammatory diseases. What is not very clear is the role played by compounds identified by UHPLC-TOF-HRMS as no assays have been performed with these molecules. In this regard, I suggest reformulating the final part of the discussion (lines 229-234).
A: We are going to the study on the isolation and identification of BF-L. 
-The "Conclusion" part needs to be rewritten as there are several errors (lines 236-238).
A: In this study, we demonstrated that leaves of Berchemia floribunda extracts (BF-L) showed anti-inflammatory effect through attenuating the generation of the pro-inflammatory mediators such as NO, iNOS, COX-2, IL-1β, IL-6 and TNF-α via the inhibition of NF-κB and MAPK signaling activation. For the first time, the anti-inflammatory effect of BF against NF-κB signaling and MAPK pathways were investigated in RAW264.7 cells, and 17 compounds were identified with UHPLC-TOF-HRMS. These findings suggest that BF-L may have great potential for the development of anti-inflammatory drug to treat acute and chronic inflammatory disorders. However, the anti-inflammatory effect of BF-L in vivo and the identification of major compound from BF-L with the anti-inflammatory effect need further studies.
-Through HPLC-TOF-HRMS analysis, the authors report a list of compounds identified but did not provide quantitative information. Please add these data. What percentage of the individual compounds are present in the analyzed matrix?
A: The objective of this study is more of an anti-inflammatory activity assessment than the compounds analysis. We are planning to conduct research on compounds analysis in the future. Please understand.  
-The ethanolic extracts were investigated only by HPLC while it would have been very interesting to also evaluate the volatile composition by GC / MS to identify other molecules bioactive able to exert the anti-inflammatory effect in a synergistic way.  Why didn't the authors take this into consideration?
A: We performed GC/MS analysis. We are going to the study on the isolation and identification of BF-L. Please understand. 

Round 2

Reviewer 1 Report

The main conclusion of this manuscript is that the extracts of Berchemia floribunda exert anti-inflammatory effects via the suppression of MAPKs/NF-κB signaling pathway. Although the data in this manuscript are very worth in the field of herbal medicines, it does not reveal the component(s) of the extracts showing anti-inflammatory activity. Therefore, the authors should delete the results of HPLC analysis from this manuscript.

Author Response

The main conclusion of this manuscript is that the extracts of BF-L exert anti-inflammatory effects via the suppression of MAPKs/NF-κB signaling pathway. Also, we performed HPLC analysis to analyze the anti-inflammatory compounds of BF-L. As a result, the 17 compounds analyzed by HPLC have been reported to exhibit anti-oxidant, anti-inflammatory, and anti-diabetes activity. The inflammation is associated with anti-oxidant and anti-diabetes. So, we included the results of HPLC analysis in this manuscript. In the future, we are going to the study on the isolation and identification of BF-L.
1. trehalose: neuroprotection
2. epigallocatechin: anti-inflammatory effects
3. neochlorogenic acid: anti-inflammatory effects
4. tryptophan: anti-inflammatory effects
5. hyperoside: anti-inflammatory effects
6. catechin: anti-inflammatory effects 
7. Luteolin 4'-glucoside: anti-inflammatory effects
8. Isorhoifolin: anti-inflammatory properties
9. Fraxetin: Antioxidant and intestinal anti-inflammatory effects
10. kaempferol 3-neohesperidoside
11. rutin: anti-inflammatory properties
12. myricitrin: antioxidant, anti-inflammatory and antifibrotic activity
13. quercitrin: anti-inflammatory properties
14. kaempferol: anti-inflammatory effects
15. pinoquercetin: anti-inflammatory effects
16. 4',5,7-trihydroxy-3,6-dimethoxyflavone: Antitumor, antioxidant and anti-inflammatory activities
17. pheophorbide a: Anti-inflammatory activity

Thank you.

Reviewer 3 Report

The authors did not refer in any way, in the manuscript or in the authors' response section, in any way to the revision fragment:

“The drawback of the study seems the phytochemical analysis limited to only one, the most active extract obtained from leaves. As the extracts of branches also displayed some anti-inflammatory activity when the fruit extract was inactive, a comparison of the phytochemical profile and the content of individual compounds in all tested extracts could be interesting from a scientific point of view. Consequently,  it could be more clearly stated which compounds could determine the high activity of one specific extract and are present in significant amounts in the active fractions or are absent in the inactive extracts. As the plant material shows great phytochemical variability, it would be more interesting than referring to literature data.“

The manuscript wasn’t improved according to the review nr. 1, only one sentence was corrected. It is impossible to accept the revision which omits the central fragment of the review.

Author Response

Thank you for your review.

 “The drawback of the study seems the phytochemical analysis limited to only one, the most active extract obtained from leaves. As the extracts of branches also displayed some anti-inflammatory activity when the fruit extract was inactive, a comparison of the phytochemical profile and the content of individual compounds in all tested extracts could be interesting from a scientific point of view. Consequently,  it could be more clearly stated which compounds could determine the high activity of one specific extract and are present in significant amounts in the active fractions or are absent in the inactive extracts. As the plant material shows great phytochemical variability, it would be more interesting than referring to literature data.“

A: NO production inhibition(%) resulted in the highest activity in the leaves. So, we performed in vitro study to evaluate anti-inflammatory effects of BF-leaves and then elucidate the potential mechanisms.

The manuscript wasn’t improved according to the review nr. 1, only one sentence was corrected. It is impossible to accept the revision which omits the central fragment of the review.

A: We modified 27-43 lines. Please check the attached file.

Reviewer 4 Report

The authors have provided the answers and the paper is now suitable for publication

Author Response

Thank you.

Round 3

Reviewer 3 Report

Unfortunately, I still consider the phytochemical part as tentative, limited, and not meeting analytical and scientific standards. The entire phytochemical discussion is based on one sentence. “Also, phytochemicals were identified as the major components (trehalose, epigallocatechin, neochlorogenic 258 acid, tryptophan, hyperoside, catechin, Luteolin 4`-glucoside, isorhoifolin, fraxetin, 259 kaempferol 3-neohesperidoside, rutin, myricitrin, quercitrin, kaempferol, pinoquercetin, 260 4',5,7-trihydroxy-3,6-dimethoxyflavone and pheophorbide a) of anti-inflammatory activity of BF-L by UHPLC-TOF-HRMS.” This sentence is an exaggeration on several levels
Only 17 from the dozens of compounds were tentatively characterized.  There are still more than 30 unidentified peaks that occur on chromatogram with high intensively, in some higher that intensity of tentatively identified 17 compounds. There is not quantitative available and without it, we are not able to predict that any of the compounds are “major compounds”. What is more, any from the unidentified compounds could be the most active one, though not characterized. 
Moreover, there is a high probability that this ubiquitous in plant parts is also present in other parts of the plants, that are not active in the used biological test. It also probable that not the compound profile, but quantity determines the differences in activity. 
What is more the sense of the sentence “Also, phytochemicals were identified as the major components […] of anti-inflammatory activity of BF-L by UHPLC-TOF-HRMS.” This is not true, because the authors did not test the anti-inflammatory activity of individual compounds. If when writing this part of the discussion, authors wanted to refer to the works of other groups, they did not do it.
I the present form, the phytochemical analysis doesn’t provide any answer, only premises, for the scientific question, what are the extract features that describe the biological activity of the active extract. But the issue isn’t discussed in the text in any way, suggesting that “ what hasn’t been achieved it the matter of fact - the part lack references. 
To my opinion, the phytochemical part, with the limited data from the study and residual and lacking conscientiousnessdiscussion, is unacceptable in the present form of the manuscript.

Author Response

The aim of this manuscript is that the extracts of BF-L exert anti-inflammatory effects via the suppression of MAPKs/NF-κB signaling pathway. So, we delated the results of HPLC analysis in this manuscript. In the future, we are going to the study on the isolation and identification of BF-L. 
